# BINNING AS A PRETEXT TASK: IMPROVING SELF-SUPERVISED LEARNING IN TABULAR DOMAINS

## ABSTRACT

The ability of deep networks to learn superior representations hinges on leveraging the proper inductive biases, considering the inherent properties of datasets. In tabular domains, it is critical to effectively handle heterogeneous features (both categorical and numerical) in a unified manner and to grasp irregular functions like piecewise constant functions. To address the challenges in the self-supervised learning framework, we propose a novel pretext task based on the classical *binning* method. The idea is straightforward: *reconstructing the bin indices* (either orders or classes) rather than the original values. This pretext task provides the encoder with an inductive bias to capture the irregular dependencies, mapping from continuous inputs to discretized bins, and mitigates the feature heterogeneity by setting all features to have category-type targets. Our empirical investigations ascertain several advantages of binning: compatibility with encoder architecture and additional modifications, standardizing all features into equal sets, grouping similar values within a feature, and providing ordering information. Comprehensive evaluations across diverse tabular datasets corroborate that our method consistently improves tabular representation learning performance for a wide range of downstream tasks. The codes are available in the supplementary material.

## 1 INTRODUCTION

Tabular datasets are ubiquitous across diverse applications from financial markets and healthcare diagnostics to e-commerce personalization and manufacturing process automation. These datasets are structured with rows representing individual samples and columns representing heterogeneous features—a combination of categorical and numerical features—and they serve as the foundation for myriad analyses. However, despite the wide applicability of tabular data, research into leveraging deep networks to harness the inherent properties of such datasets is still in its nascent stage. In contrast, tree-based machine learning algorithms like XGBoost (Chen & Guestrin, 2016) and CatBoost (Prokhorenkova et al., 2018) have consistently demonstrated prowess in discerning the nuances of tabular domains, outperforming deep networks even those with a larger model capacity and specialized modules (Arik & Pfister, 2021; Gorishniy et al., 2021; Grinsztajn et al., 2022; Rubachev et al., 2022). The consistent edge held by tree models fuels the exploration of how their advantageous biases can be adapted for deep networks.

Recently, the quest to boost the performance of deep networks on tabular data has gained momentum. A fundamental challenge is the inherent heterogeneity of tabular datasets, encompassing both categorical and numerical features (Popov et al., 2019; Borisov et al., 2022; Yan et al., 2023). To mitigate the feature discrepancies in deep networks, previous studies proposed using an additional module like a feature tokenizer (Gorishniy et al., 2021) and an abstract layer (Chen et al., 2022). Concurrently, some research has explored ways to infuse the proven strengths of tree-based models into deep networks. For instance, Grinsztajn et al. (2022) observed that deep networks tend to prefer overly smooth solutions and struggle with modeling irregularities like piecewise constant functions, in contrast to the tree-based models. To address this challenge, Gorishniy et al. (2022) introduced a novel approach combining piecewise linear encoding during preprocessing and periodic activation functions. Although these advancements have led to enhanced performance on several tabular data problems, they have predominantly been explored within a supervised learning framework.

To expand the success of deep networks on tabular domain to unsupervised representation learning, we propose a novel pretext task based on the classical *binning* method for auto-encoding-based self-supervised learning (SSL). Our approach is straightforward: *reconstructing bin indices* rather than reconstructing the raw values, as illustrated in Figure 1. Once numerical features are discretized into bins based on the quantiles of the training dataset, we optimize the encoder and decoder networks to accurately predict the bin indices given original inputs. By setting the discretized bins as targets for the pretext task, we can employ the inductive bias of capturing the irregular functions and mitigating the discrepancy between features. The binning procedure allows grouping the nearby samples based on the distribution of the training dataset, so the learned representations should be robust to the minor errors that can yield spurious patterns. It also facilitates standardizing all features into equal sets, thereby preventing any uninformative features from dominating during SSL. Furthermore, our

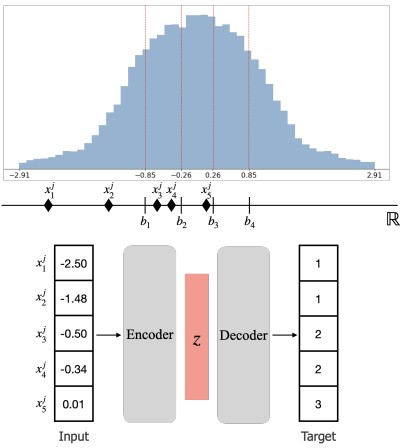

Figure 1: Binning as a pretext task. Bins are determined based on the distribution of the training dataset for each feature. The inputs are passed into the encoder network, then the decoder network predicts the bin indices which can be ordinal when the pretext task is the regression or nominal when the pretext task is the classification.

approach is compatible with any other modifications, including the choice of deep architectures and input transformation functions.

Based on the extensive experiments on 25 public datasets, we found that the binning task consistently improves the SSL performance on diverse downstream tasks, even though we simply changed the targets during SSL from the continuous to the discretized bins. The performance is also comparable with the supervised counterparts, including state-of-the-art deep networks and tree-based machine learning algorithms. Finally, we found that the binning task can be not only an effective objective function for fully unsupervised learning but also beneficial as the pretraining strategy.

Our main contributions can be summarized as follows. First, we suggest binning as a new pretext task for SSL in tabular domains, compatible with any modifications. Second, we conduct extensive experiments on 25 public tabular datasets focusing on the various input transformation methods and SSL objectives. Finally, we empirically found that the binning task not only results in better representations but also provides good initial weights for fine-tuning in various datasets and downstream tasks. The codes are available in the supplementary material.

## 2 RELATED WORKS

**Tabular deep learning:** In recent years, there has been a large number of deep learning research on a tabular domain: developing new deep architectures (Popov et al., 2019; Badirli et al., 2020; Huang et al., 2020; Wang et al., 2021; Arik & Pfister, 2021; Gorishniy et al., 2021; Ucar et al., 2021; Chen et al., 2022; Zhu et al., 2023; Kotelnikov et al., 2023; Chen et al., 2023); or representing the heterogeneous nature of tabular features as the graphs (Yan et al., 2023); or adopting new activation function (Gorishniy et al., 2022). Despite these advancements, ensembles of decision trees, such as GBDTs (Gradient Boosting Decision Trees), continue to serve as competitive baselines (Arik & Pfister, 2021; Gorishniy et al., 2021; Grinsztajn et al., 2022; Rubachev et al., 2022). In this paper, our goal is to suggest a new pretext task for self-supervised learning in tabular domains, so we focus on architectures directly inspired by classic deep models, in particular MLPs and FT-Transformers (Gorishniy et al., 2021). Additionally, we provide comparisons with several deep learning-based approaches and tree-based machine learning algorithms.

**Self-supervised learning in tabular domains:** Self-supervised learning (SSL) aims to learn desirable representations without making use of annotation information. Recently, contrastive learning and auto-encoding have been two major choices in the tabular domain. Contrastive learning aims to

model the similarity between two or more augmented views from the same sample, corresponding to the positive samples, and the dissimilarity between other samples, corresponding to the negative samples. Bahri et al. (2021); Ucar et al. (2021) have optimized contrastive loss after defining the positive and negative samples based on the data augmentation function, such as masking or cropping in feature dimension. Auto-encoding aims to reconstruct the original sample given its corrupted observation (Vincent et al., 2008). Compared to contrastive learning, auto-encoders can handle a mix of data types which can be beneficial for tasks involving heterogeneous datasets, like tabular data. Yoon et al. (2020); Huang et al. (2020); Majmundar et al. (2022) adopted the auto-encoding methods optimizing the reconstruction loss with or without the additional losses, such as corruption detection. In this study, we suggest a new SSL pretext task based on the auto-encoding approach.

## 3 PRELIMINARIES: AUTO-ENCODING-BASED SELF-SUPERVISED LEARNING IN TABULAR DOMAINS

Autoencoding is a classical method for learning representations with a variety of use cases. Numerous methods have been suggested to generalize denoising autoencoders (Vincent et al., 2008) in the context of SSL, which aim to learn representations by reconstructing original signals from corrupted samples. In this section, we delve into the auto-encoding-based self-supervised learning framework in tabular domains focusing on two factors: transformation methods to tabular inputs and the objective functions in the auto-encoding-based SSL framework.

**Input transformation:** To ensure the encoder network does not simply learn an identity function, we employ transformation functions on the input that preserve the label-related information. For tabular datasets, only a few transformation functions have been suggested like masking (Yoon et al., 2020; Ucar et al., 2021; Majmundar et al., 2022) as illustrated in Figure 2 because all individual values can play a key role in determining the semantics and small changes can affect the context. Given a sample $x_i \in \mathbb{R}^d$ in dataset $\mathcal{D}$ where $d$ is the number of features, $i \in [1, N]$, and $N$ is the batch size, we randomly generate the masking vector $m_i$ with the same size of $x_i$. Each element of the masking vector $m_i$ is independently sampled from a Bernoulli distribution with probability $p_m \in [0, 1]$. To replace the masked values, the replacing vector $\bar{x}_i$ should be defined. In this study, we utilize two methods suggested in the previous studies (Yoon et al., 2020; Ucar et al., 2021; Majmundar et al., 2022).

- Constant (Figure 2a): $\bar{x}_{i,k}$ is set as the pre-determined constant value for all $i$. In this study, we use the average for each feature $k$ in the training dataset.
- Random (Figure 2b): $\bar{x}_{i,k}$ is sampled from the other in-batch samples for a given feature. In other words, to replace the $k$-th feature of the $i$-th sample in the batch, we use the $k$-th feature of the $i'$-th sample in the same batch, and $i'$ is sampled from the uniform distribution $\mathcal{U}\left(\frac{1}{N}\right)$.

Finally, the corrupted sample $\tilde{x}_i$ is formulated as $\tilde{x}_i = (\mathbf{1} - m_i) \odot x_i + m_i \odot \bar{x}_i$ where $\mathbf{1}$ is all-ones vector with the same size of $x_i$. The transformation procedure is stochastic and it provides randomness during training. When $p_m = 0$, $m_i$ becomes the zero matrix, and the uncorrupted input $\tilde{x}_i = x_i$ is used for training.

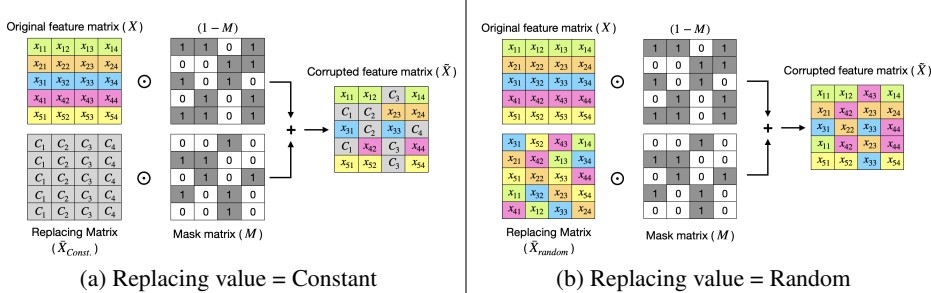

(a) Replacing value = Constant  (b) Replacing value = Random

Figure 2: An illustration of two methods to generate the replacing vectors for masked features.

**SSL objectives:** Following the convention of SSL, the encoder $f_e$ first transforms the corrupted sample $\tilde{x}_i$ to a representation $z_i$, then the decoder $f_d$ will be introduced to learn the informative

representation by optimizing the unsupervised loss $\mathcal{L}$. We can leverage which representation should be learned by introducing the specific pretext task. As a baseline, we consider two pretext tasks used in Yoon et al. (2020); Huang et al. (2020); Majmundar et al. (2022).

- Reconstructing the original values: One common approach is to reconstruct uncorrupted samples from their corrupted counterparts (Vincent et al., 2008). In this setup, the encoder attempts to impute the masked features by leveraging the correlations present in the non-masked features. The learned representations will involve the semantic-level information that is invariant to corruption. To this end, the decoder network is defined as $f_d^{\text{recon}} : Z \to \hat{X}$, and the corresponding loss is formulated as $\mathcal{L}_{\text{ValueRecon}} := \frac{1}{N} \sum_{i=1}^{N} ||x_i - f_d^{\text{recon}}(z_i)||_2^2$.
- Detecting the masked features: An auxiliary task that can facilitate the pretext task of reconstruction is predicting which features have been masked during the corruption process of the input sample (Yoon et al., 2020). In this setup, the encoder attempts to leverage the inconsistency between feature values to identify the masked features, resulting in learned representations that capture abnormal patterns for a given input. Specifically, the method employs a binary cross-entropy loss, which can be formulated as $\mathcal{L}_{\text{MaskXent}} := -\frac{1}{N} \sum_{i=1}^{N} m_i \log f_d^{\text{mask}}(z_i) + (\mathbf{1} - m_i) \log (\mathbf{1} - f_d^{\text{mask}}(z_i))$ where the decoder network is defined as $f_d^{\text{mask}} : Z \to \hat{M}$.

We can optimize several loss functions simultaneously if we train several decoders that utilize $z$ as the inputs. For example, Yoon et al. (2020) utilized the weighted sum of $\mathcal{L}_{\text{ValueRecon}}$ and $\mathcal{L}_{\text{MaskXent}}$.

## 4 METHODS: BINNING AS A PRETEXT TASK FOR TABULAR SSL

Binning is a classical data preprocessing technique that quantizes a given numerical feature $x^j \in \mathbb{R}^{|\mathcal{D}|}$ into $T$ discrete intervals, known as *bins* $B_t^j = [b_{t-1}^j, b_t^j)$ where $t \in [1, T]$ and $b_t^j \in \mathbb{R}$ is the bin boundaries. Binning is effective in transforming continuous features into discrete ones, mitigating minor errors in datasets like noise and outliers, and making the data distribution more manageable (Dougherty et al., 1995; Han et al., 2022).

In this study, we implement binning to establish targets for auto-encoding-based SSL. We anticipate the representations will be robust to the minor input variation in the same bins. Also, the deep networks can capture the irregularities akin to the decision-making process of tree-based models, which assign discrete leaves to each continuous sample because the pretext task corresponds to mapping continuous inputs to discretized bins. Additionally, the binning approach helps mitigate feature heterogeneity by treating the targets for all features as the same category type during SSL.

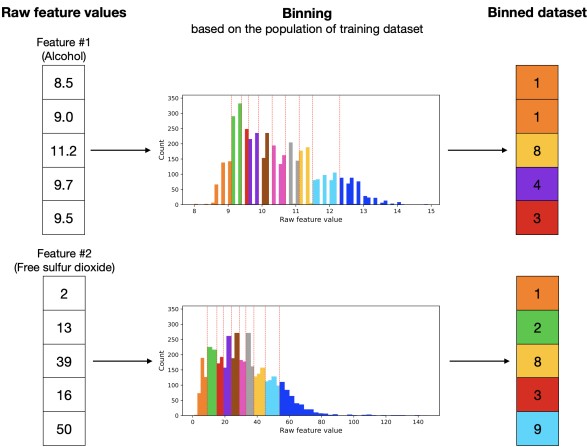

Figure 3: An example of binning (Dataset: Wine Quality (Cortez et al., 2009)). In the example, we set $T$ as 10. For each feature, we implement the binning to include the same number of observations based on the training dataset. Finally, we use the binning indices as the targets for auto-encoding-based SSL. When we regard the bin indices as the classes without order information, the binning indices are converted into the one-hot vectors.

The binning procedure is described in Figure 3. We first determine the number of bins $T$ as the design parameter. Then, we split the value range into the disjoint set of $T$ intervals, $\left\{ B_1^j, \ldots, B_T^j \right\}$, considering the number of observations in the training dataset $\mathcal{D}_{\text{train}}$ for each $j$-th feature $x^j$. Specifically, the bin boundaries $b_t^j$ are determined according to the quantiles of $\frac{t}{T}$. (Alternative binning strategies are also discussed in Supplementary D.1.) When the number of unique values for $x^j$ in the training dataset is less than $T$, each distinct value is assigned its own bin. Finally, we place each numerical feature $x_i^j$ into the bin $B_t^j$, and we substitute the original values with the corresponding bin indices $t_i^j \in [1, T]$. Thus, we use the grouped ranks (or classes) instead of the raw values. We call the binned dataset as $X_{\text{Bin}}$.

The bin index of $i$-th sample and $j$-th feature, $t_i^j$, can be expressed as ordinal values or nominal classes. When we utilize the bin indices as ordinal values, we set the pretext task as reconstructing the bin indices based on the continuous inputs, and the corresponding *BinRecon loss* is defined as

$$\mathcal{L}_{\text{BinRecon}} := \frac{1}{N} \sum_{i=1}^N \left\| t_i - f_d^{\text{BinRecon}}(z_i) \right\|_2^2 \text{ where } f_d^{\text{BinRecon}} : Z \to \hat{X}_{\text{Bin}}. \tag{1}$$

When we utilize the bin indices as nominal classes, we convert the bin index $t_i^j$ into the one-hot vector $\mathbf{u}_i^j = [u_1, u_2, \ldots, u_T]$ where $u_v = 1$ when $v = t_i^j$ and $u_v = 0$ otherwise. Then, we set the pretext task as predicting the bin indices as classes by optimizing the *BinXent loss*, defined as the average of binary cross-entropy loss for each feature.

$$\mathcal{L}_{\text{BinXent}} := -\frac{1}{Nd} \sum_{i=1}^N \sum_{j=1}^d \mathbf{u}_i^j \log f_d^{\text{BinXent}}(z_i^j) + (\mathbf{1} - \mathbf{u}_i^j) \log \left( \mathbf{1} - f_d^{\text{BinXent}}(z_i^j) \right) \tag{2}$$

In this case, the predictions for each sample should be in $\mathbb{R}^{d \times T}$. As a simple implementation, we add the 1x1 convolutional layer at the end of $f_d^{\text{BinXent}}(\cdot) : Z \to \hat{U}$ where $U \in \mathbb{R}^{N \times d \times T}$ represents the one-hot encoded binned dataset.

We outline the benefits of utilizing the binning task in SSL as follows. Empirical evidence on how each item is advantageous for tabular data problems will be provided in subsequent sections.

- *Compatibility with any other modifications*: The binning task is agnostic to modifications such as changes in encoder architecture, input transformation functions, and additional objectives. Therefore, it can be utilized independently or in conjunction with other options. (Section 5.1, 5.2)

- *Standardizing all features into equal sets*: After binning, all features lie on the uniform distribution with identical elements. Unlike the conventional normalization schemes, it largely simplifies the dataset to include only $T$ distinct values, and this ensures all features become equal sets, thereby preventing any uninformative features from dominating during training. (Section 6.1)

- *Grouping similar values in each feature*: Binning clusters the nearby values in each feature and eliminates the other information except the bin index. Deep networks can identify nearby samples in a distribution as similar, independent of their magnitude. (Section 6.1)

- *Ordering in BinRecon loss*: BinRecon loss utilizes the grouped rank information only while eliminating the raw value information. This ensures that the encoder network learns the ordering information, regardless of the magnitude of the values. (Section 6.1)

Overall, we implement SSL as follows. First, tabular inputs undergo a transformation that retains their semantic information. Then, the encoder network $f_e$ takes the transformed input $\tilde{x}$ and produces the representation $z$, and the decoder network $f_d$ models the representation $z$ to the target $\hat{y}_{\text{SSL}}$ depending on the choice of pretext task. In this study, we consider four types of pretext tasks and the corresponding losses are ValueRecon, MaskXent, BinRecon, and BinXent. Once SSL is finished, the learned representations $z$ are evaluated based on linear probing.

## 5 EXPERIMENTS

In this section, we evaluate the effectiveness of binning as a pretext task across 25 public tabular datasets encompassing a range of data sizes and task types. Dataset details are provided in Supplementary A. For all datasets, we apply standardization for numerical features and labels for evaluating the regression tasks.

As the encoder network $f_e$, we mainly utilize the MLP networks without a special module. Note that a larger or more complex network does not guarantee better performance in tabular datasets (Gorishniy et al., 2021; Rubachev et al., 2022; Grinsztajn et al., 2022; Gorishniy et al., 2022). To determine the depth and width of $f_e$, we identify the optimal configuration based on validation performance in the supervised setup, i.e., only the encoder with a linear head is trained with the supervised loss, ensuring the unsupervised nature of our framework. Then, the decoder $f_d$ mirrors $f_e$ in architecture. Consequently, all cases for each dataset have been trained on the same architecture and optimization setups. A detailed description is provided in Supplementary B. For a given network and dataset, we also investigate the masking probability $p_m \in \{0.1, 0.2, 0.3, 0.4, 0.5, 0.6, 0.7, 0.8, 0.9\}$ and the number of bins $T \in \{2, 5, 10, 20, 50, 100\}$. Then, we found the optimal configuration based on validation performance on each downstream task. After SSL, we evaluate the representations based on linear probing 10 times with different random seeds, and an average is reported. We evaluate the representation quality based on accuracy for classification tasks and RMSE for regression tasks. The full results with standard deviation are also available in Supplementary C. All experiments are conducted on a single NVIDIA GeForce RTX 3090.

## 5.1 Comparing SSL models

We first compare a series of auto-encoding-based SSL methods utilizing the same MLP networks for each dataset. To identify the compatibility of the binning task with other transformation functions, we include the cases optimizing BinRecon loss with masking transformation. Finally, we experiment with four cases to validate our methodology; optimizing BinXent, treating bins as nominal classes; optimizing BinRecon, treating bins as ordinal values without any augmentation; optimizing BinRecon with masking as constant values; and optimizing BinRecon with masking as random values. The results for each type of downstream task are summarized in Table 1, and four rows at the bottom correspond to our methods.

**Binary classification:** First, we compare the performance of eight datasets whose downstream task is binary classification. For all cases, except PH dataset, binning as a pretext task shows the best performance against the other methods. In particular, BinRecon without input transformation performs best for three datasets, and the average rank is 2.375 among 11 SSL methods. Interestingly, for HI dataset, the performance has been improved from 0.651 to 0.687(+3.6%), 0.653 to 0.672(+1.9%), and 0.661 to 0.682(+2.1%) when we simply change the target for reconstruction loss from the raw values to bin indices. Similar patterns are often observed in other datasets. These results indicate that learning irregular functions (from continuous to discrete) is more beneficial than learning smooth functions (from continuous to continuous) in tabular representation learning.

**Multiclass classification:** Next, we investigate nine datasets whose downstream task is multiclass classification. Unlike the binary classification tasks, we observe that optimizing BinRecon loss with masking consistently leads to additional improvements compared to the cases without masking, and optimizing BinXent does not work well. These results indicate that the order information is important for multiclass classification and BinRecon can effectively manipulate them. Further discussion will be provided in Section 6. When we compare the results of MNIST and p-MNIST, the effectiveness of the binning task, especially for the tabular datasets, becomes clear. Because MNIST is a simply flattened dataset of handwritten images, the locality should exist and it is not a general property of the tabular datasets. To make the dataset more tabular-like, we permute the values and call them p-MNIST. We observe that most SSL methods achieve good performance for MNIST (worst 0.793, best 0.966), while the performance degrades quite a lot for p-MNIST (worst 0.554, best 0.934). On the other hand, we found that BinRecon consistently achieves a great performance in both datasets (MNIST: 0.964 to 0.981, p-MNIST: 0.950 to 0.971). Thus, the binning task could lead to learning good representations even when the inter-feature dependency is rarely quantified.

**Regression:** Finally, we test eight datasets whose downstream task is the regression. Since the evaluation metric is RMSE, lower values correspond to better-performing cases. Again, the binning task consistently improves the SSL performance, and BinRecon with masking as the random values shows the best performance with an average rank of 1.625. Compared to other downstream tasks, regression tasks exhibit the most significant improvements with the binning pretext task. For in-

Table 1: Linear evaluation results for various SSL methods. For each method, we also determine the performance rankings for each dataset, and the average ranks are also provided in the last column.

(a) Binary classification (Metric: Accuracy)

| Masking | Replacing value | SSL Objective(s) | CH | HI | AD | BM | PH | OS | CS | PO | Average Rank |
|---|---|---|---|---|---|---|---|---|---|---|---|
| FALSE | - | ValueRecon | 0.810 | 0.651 | 0.837 | 0.899 | 0.728 | 0.883 | 0.709 | 0.851 | 7.625 |
| TRUE | Const. | MaskXent | 0.807 | 0.672 | 0.836 | 0.899 | 0.715 | 0.893 | 0.708 | 0.845 | 7.500 |
| TRUE | Const. | ValueRecon | 0.810 | 0.653 | 0.839 | 0.900 | 0.734 | 0.884 | 0.718 | 0.849 | 6.000 |
| TRUE | Const. | MaskXent+ValueRecon | 0.817 | 0.669 | 0.835 | 0.900 | 0.724 | 0.877 | 0.706 | 0.837 | 8.000 |
| TRUE | Random | MaskXent | 0.814 | 0.681 | 0.843 | **0.901** | 0.710 | 0.883 | 0.706 | 0.853 | 6.000 |
| TRUE | Random | ValueRecon | 0.811 | 0.661 | 0.838 | 0.898 | **0.738** | 0.885 | 0.714 | 0.842 | 6.875 |
| TRUE | Random | MaskXent+ValueRecon | 0.804 | 0.647 | 0.826 | 0.899 | 0.715 | 0.879 | 0.713 | 0.861 | 8.375 |
| FALSE | - | BinXent | 0.817 | 0.683 | 0.845 | **0.901** | 0.732 | 0.886 | **0.738** | 0.851 | 3.250 |
| FALSE | - | BinRecon | **0.823** | **0.687** | 0.840 | 0.900 | 0.737 | 0.889 | 0.724 | **0.865** | **2.375** |
| TRUE | Const. | BinRecon | 0.820 | 0.672 | 0.843 | 0.899 | 0.730 | **0.896** | 0.718 | 0.858 | 3.625 |
| TRUE | Random | BinRecon | 0.819 | 0.682 | **0.846** | 0.898 | 0.735 | 0.894 | 0.718 | 0.858 | 3.500 |

(b) Multiclass classification (Metric: Accuracy)

| Masking | Replacing value | SSL Objective(s) | CO | OT | GE | VO | WQ | AL | HE | MNIST | p-MNIST | Average Rank |
|---|---|---|---|---|---|---|---|---|---|---|---|---|
| FALSE | - | ValueRecon | 0.769 | 0.776 | 0.527 | 0.619 | 0.568 | 0.931 | 0.353 | 0.965 | 0.928 | 6.333 |
| TRUE | Const. | MaskXent | 0.784 | 0.777 | 0.518 | 0.545 | 0.547 | 0.909 | 0.341 | 0.793 | 0.554 | 9.333 |
| TRUE | Const. | ValueRecon | 0.783 | 0.791 | 0.557 | 0.622 | 0.586 | 0.931 | 0.354 | 0.966 | 0.925 | 4.111 |
| TRUE | Const. | MaskXent+ValueRecon | 0.750 | 0.774 | 0.519 | 0.610 | 0.571 | 0.931 | 0.360 | 0.941 | 0.907 | 7.444 |
| TRUE | Random | MaskXent | 0.763 | 0.791 | 0.555 | 0.549 | 0.544 | 0.925 | 0.336 | 0.945 | 0.817 | 8.000 |
| TRUE | Random | ValueRecon | 0.761 | 0.782 | 0.538 | 0.625 | 0.573 | 0.930 | 0.357 | 0.956 | 0.934 | 5.556 |
| TRUE | Random | MaskXent+ValueRecon | 0.769 | 0.779 | 0.521 | 0.564 | 0.519 | 0.925 | 0.353 | 0.945 | 0.906 | 8.333 |
| FALSE | - | BinXent | 0.742 | 0.781 | 0.517 | 0.600 | 0.565 | 0.903 | 0.354 | 0.956 | 0.908 | 8.333 |
| FALSE | - | BinRecon | 0.784 | 0.783 | 0.544 | 0.625 | **0.592** | 0.935 | 0.357 | 0.964 | 0.950 | 3.556 |
| TRUE | Const. | BinRecon | 0.812 | 0.792 | 0.559 | 0.647 | 0.581 | 0.943 | 0.359 | 0.974 | 0.964 | 2.222 |
| TRUE | Random | BinRecon | **0.814** | **0.794** | **0.580** | **0.655** | 0.574 | **0.949** | **0.365** | **0.981** | **0.971** | **1.333** |

(c) Regression (Metric: RMSE)

| Masking | Replacing value | SSL Objective(s) | CA | HO | FI | MI | KI | CPU | DIA | EL | Average Rank |
|---|---|---|---|---|---|---|---|---|---|---|---|
| FALSE | - | ValueRecon | 0.749 | 4.241 | 13900.720 | 0.784 | 0.163 | 3.876 | 1016.641 | 0.399 | 8.625 |
| TRUE | Const. | MaskXent | 0.709 | 4.548 | 13473.750 | 0.788 | 0.185 | 4.475 | 1259.744 | 0.396 | 8.875 |
| TRUE | Const. | ValueRecon | 0.693 | 4.086 | 13518.683 | 0.778 | 0.160 | 3.728 | 952.444 | 0.394 | 5.000 |
| TRUE | Const. | MaskXent+ValueRecon | 0.700 | 4.157 | 13915.875 | 0.775 | 0.174 | 5.644 | 2797.034 | 0.398 | 8.750 |
| TRUE | Random | MaskXent | 0.677 | 4.297 | 13826.641 | 0.782 | 0.176 | 3.951 | 1358.135 | 0.388 | 7.875 |
| TRUE | Random | ValueRecon | 0.713 | 4.127 | 13668.988 | 0.777 | 0.162 | 3.760 | 986.306 | 0.396 | 6.500 |
| TRUE | Random | MaskXent+ValueRecon | 0.701 | 4.136 | 14107.645 | 0.780 | 0.166 | 4.506 | 1917.875 | 0.397 | 8.750 |
| FALSE | - | BinXent | 0.690 | 4.116 | **13038.762** | 0.776 | 0.170 | 3.717 | 1207.923 | 0.383 | 4.875 |
| FALSE | - | BinRecon | 0.622 | 3.766 | 13453.309 | **0.767** | **0.158** | 3.208 | 897.645 | 0.370 | 2.250 |
| TRUE | Const. | BinRecon | 0.634 | 3.765 | 13208.133 | 0.773 | **0.158** | **3.156** | 957.801 | 0.371 | 2.375 |
| TRUE | Random | BinRecon | **0.619** | **3.703** | 13075.474 | 0.773 | 0.160 | 3.183 | **870.283** | **0.368** | **1.625** |

stance, when comparing our method with the best baselines, we observed improvements of 10.27% for HO dataset, 8.63% for DIA dataset, and 8.57% for CA dataset.

## 5.2 COMPARISON WITH THE SUPERVISED COUNTERPARTS

We observed that the binning as a pretext task consistently improves the SSL performance across the various tabular datasets and the downstream tasks. In this section, we compare our method with the supervised counterparts, consisting of the encoder and linear head. For supervised baselines, we employ the encoder networks with random weights (*Baseline-1*) or trained from scratch with a supervised objective (*Baseline-2*). For our methods, encoder networks are first trained with BinXent or BinRecon loss, and then the learned representations are evaluated through linear probing (*Ours-1*) or fine-tuning (*Ours-2*). To investigate the effectiveness of the binning task with different encoder architectures, we also experiment with the FT-Transformer (Gorishniy et al., 2021), a simple adaptation of the Transformer architecture for tabular data without additional hyperparameter tuning.

The results are provided in Table 2 and Table 8 in supplementary material. For most datasets, we found that our methods in unsupervised setups (*Ours-1*) achieve comparable performance with the supervised baselines. After fine-tuning (*Ours-2*), pre-trained models on the binning task frequently outperform the supervised baselines. Binning-based models show the best performance for all datasets regardless of the choice of encoder architecture, with the exception of OT with FT-Transformer. Overall, we found that SSL based on the binning task can be an effective method to learn both the good representations and the initial weights for fine-tuning.

Table 2: Comparison with supervised baselines. We compare the downstream task performance under several scenarios: (1) Baseline-1, where the encoder is randomly initialized; (2) Baseline-2, where the encoder is trained by optimizing the supervised loss; (3) Ours-1, where the encoder is trained based on the binning task only; and (4) Ours-2, where the encoder is fine-tuned after the pre-training on binning tasks. (Notation: ↑ corresponds to accuracy, ↓ corresponds to RMSE)

| Training method | CH↑ | AD↑ | PH↑ | OS↑ | CO↑ | OT↑ | GE↑ | VO↑ | HE↑ | MNIST↑ | CA↓ | HO↓ | FI↓ | EL↓ |
|---|---|---|---|---|---|---|---|---|---|---|---|---|---|---|
| *Encoder = MLP* | | | | | | | | | | | | | | |
| Baseline-1 | 0.796 | 0.820 | 0.683 | 0.873 | 0.729 | 0.766 | 0.467 | 0.547 | 0.311 | 0.896 | 0.854 | 4.700 | 14241.610 | 0.400 |
| Baseline-2 | 0.836 | 0.849 | 0.724 | 0.895 | 0.968 | 0.810 | 0.659 | 0.694 | 0.378 | 0.983 | 0.513 | 3.146 | 10086.080 | 0.354 |
| Ours-1 | 0.823 | 0.846 | 0.736 | **0.896** | 0.814 | 0.794 | 0.580 | 0.655 | 0.365 | 0.981 | 0.619 | 3.703 | 13038.762 | 0.368 |
| Ours-2 | **0.841** | **0.854** | **0.738** | 0.895 | **0.969** | **0.814** | **0.675** | **0.724** | **0.385** | **0.986** | **0.502** | **3.026** | **9963.609** | **0.350** |
| *Encoder = FT-Transformer* | | | | | | | | | | | | | | |
| Baseline-1 | 0.818 | 0.828 | 0.694 | 0.866 | 0.730 | 0.705 | 0.509 | 0.544 | 0.311 | 0.550 | 0.690 | 4.107 | 16128.694 | 0.394 |
| Baseline-2 | 0.824 | 0.837 | 0.724 | 0.884 | 0.970 | **0.794** | 0.664 | 0.704 | 0.338 | 0.966 | 0.487 | 3.319 | 10206.127 | 0.350 |
| Ours-1 | **0.836** | **0.853** | 0.725 | **0.887** | 0.762 | 0.780 | 0.554 | 0.614 | **0.364** | 0.931 | 0.549 | 3.570 | 14557.626 | 0.371 |
| Ours-2 | 0.834 | 0.839 | **0.734** | 0.882 | **0.971** | 0.793 | **0.698** | **0.720** | 0.342 | **0.978** | **0.477** | **3.173** | **9936.115** | **0.343** |

Table 3: Comparison with the tree-based machine learning algorithms and the recent deep learning methods including state-of-the-art models.

| Training network and method | Supervised | CH↑ | HI↑ | AD↑ | CO↑ | OT↑ | GE↑ | AL↑ | HE↑ | MNIST↑ | CA↓ | HO↓ | MI↓ |
|---|---|---|---|---|---|---|---|---|---|---|---|---|---|---|
| ***Tree-based machine learning algorithms*** | | | | | | | | | | | | | | |
| XGBoost | Yes | 0.859 | 0.726 | **0.875** | 0.969 | **0.827** | 0.683 | 0.924 | 0.348 | 0.980 | 0.434 | 3.152 | 0.742 |
| CatBoost | Yes | 0.861 | 0.727 | 0.873 | 0.967 | 0.825 | 0.692 | 0.948 | 0.386 | 0.980 | **0.430** | 3.093 | **0.741** |
| ***Recent deep learning methods*** | | | | | | | | | | | | | | |
| TabNet (Arik & Pfister, 2021; Gorishniy et al., 2021) | Yes | - | 0.719 | 0.850 | 0.957 | - | - | 0.954 | 0.378 | - | 0.510 | - | 0.751 |
| NODE (Popov et al., 2019; Gorishniy et al., 2021) | Yes | - | 0.726 | 0.858 | 0.958 | - | - | 0.918 | 0.359 | - | 0.464 | - | 0.745 |
| GrowNet (Badirli et al., 2020; Gorishniy et al., 2021) | Yes | - | 0.722 | 0.857 | - | - | - | - | - | - | 0.487 | - | 0.751 |
| DCN V2 (Wang et al., 2021; Gorishniy et al., 2021) | Yes | - | 0.723 | 0.853 | 0.965 | - | - | 0.955 | - | - | 0.484 | - | 0.749 |
| PLR (MLP-Ensemble) (Gorishniy et al., 2022) | Yes | 0.857 | 0.728 | 0.870 | 0.970 | 0.819 | 0.674 | - | - | - | 0.467 | 3.050 | 0.746 |
| PLR (FT-T-Ensemble) (Gorishniy et al., 2022) | Yes | **0.863** | 0.730 | 0.870 | **0.972** | 0.814 | 0.646 | - | - | - | 0.464 | 3.162 | 0.746 |
| T2G-Former (Yan et al., 2023) | Yes | **0.863** | **0.734** | 0.860 | - | 0.819 | 0.656 | - | 0.391 | - | 0.455 | 3.138 | - |
| VIME (Yoon et al., 2020) | No | - | - | - | - | - | - | - | - | 0.958 | - | - | - |
| SubTab (Ucar et al., 2021) | No | - | - | - | - | - | - | - | - | 0.979 | - | - | - |
| Ours (MLP) | No | 0.823 | 0.687 | 0.846 | 0.814 | 0.794 | 0.580 | 0.949 | 0.365 | 0.981 | 0.619 | 3.703 | 0.767 |
| Ours (MLP+Finetuning) | Yes | 0.841 | 0.716 | 0.854 | 0.969 | 0.814 | 0.675 | **0.963** | 0.385 | **0.986** | 0.502 | **3.026** | 0.753 |
| Ours (FT-T) | No | 0.836 | 0.670 | 0.853 | 0.762 | 0.780 | 0.554 | 0.930 | 0.364 | 0.931 | 0.549 | 3.570 | 0.770 |
| Ours (FT-T+Finetuning) | Yes | 0.835 | 0.700 | 0.839 | 0.971 | 0.793 | **0.698** | 0.961 | 0.342 | 0.978 | 0.477 | 3.173 | 0.752 |

## 5.3 COMPARISON WITH THE TREE-BASED AND THE STATE-OF-THE-ART METHODS

In addition to the supervised counterparts, we also compare our methods with the state-of-the-art deep networks and the tree-based machine learning algorithms, such as XGBoost (Chen & Guestrin, 2016) and CatBoost (Prokhorenkova et al., 2018). To minimize the ambiguity from the choice of random seeds and hyperparameters, we directly reference the reported performances in the papers, so the model capacity and the optimization strategies would be different.

The results are provided in Table 3. Remarkably, even without utilizing annotation information during pretraining, our approach based on a simple MLP yields performance comparable to most existing supervised methods. For the MNIST dataset, our method accomplishes the best performance in the unsupervised setup. Among 12 datasets, pretraining on the binning task has achieved the best performance for four datasets, with particularly notable improvements in multiclass classification tasks. The consistently superior performance of our method supports the wide usability of binning as the pretext task in tabular domains.

## 6 DISCUSSION

### 6.1 ABLATION STUDY: WHAT IS THE MOST IMPORTANT FACTOR FOR BINNING?

In this section, we scrutinize the individual contributions of the components of binning, detailed in Section 4. Specifically, we examine the roles of discerning the order of samples within each feature, standardizing all features into equal sets, and grouping similar values. BinRecon encapsulates all three elements while ValueRecon disregards them completely. To dissect the influence of each factor, we systematically eliminate them one by one from the BinRecon loss as follows.

- Ordering: We shuffle the bin indices with different random seeds for each feature.
- Standardizing into equal sets: We replace the raw values with the averages for each bin, instead of bin indices. Then, each feature includes different elements in different ranges.

- Grouping: We set $T^j = |\mathcal{D}^j_{\text{train}}|$ for every feature. In this case, each unique value corresponds to an individual bin, and only the order information remains.

The results are summarized in Table 4. Because the performance range varies depending on the datasets, we report the dataset count and the relative performance improvement/deterioration against the case optimizing BinRecon loss without input transformation corresponding to satisfying all three factors. We do not include the unchanged cases where the performance

Table 4: Ablation test results on individual components of binning.

| Ordering | Standardizing | Grouping | Improved | Deteriorated |
|---|---|---|---|---|
| Yes | Yes | Yes | - (*Baseline*) | - (*Baseline*) |
| No | Yes | Yes | 1 (+4.70%) | 12 (− 4.04%) |
| Yes | No | Yes | 1 (+5.21%) | 15 (− 6.76%) |
| Yes | No | No | - | 23 (−25.29%) |
| No | No | No | - | 18 (− 5.99%) |

change is less than 1%. Obviously, eliminating the grouping factor shows the largest performance degradation, averaging a 25.29% decrease in 23 datasets among 25. This decline is much steeper than the effect of eliminating all three factors, which only decreased performance by 5.99% in 18 datasets. From these observations, we infer that the grouping factor is most critical for the successful implementation of binning.

## 6.2 DEPENDENCY BETWEEN THE NUMBER OF BINS AND DOWNSTREAM TASK PERFORMANCE

In this section, we investigate the relationship between the number of bins and downstream task performance for BinXent and BinRecon without input transformation. Because the performance range is quite different between the datasets, we normalize the performance with the best and worst cases for each dataset. Thus, the best-performing case corresponds to 1, and the worst-performing case corresponds to 0. As shown in Figure 4, there is no clear relationship between the number of bins and performance (Pearson correlation $\rho^2 = 0.01$, Kendall rank correlation $\tau = 0.16$ for BinXent, $\rho^2 = 0.04$, $\tau = 0.27$ for BinRecon), except that the number of bins should be not too small, but larger is not always better. This

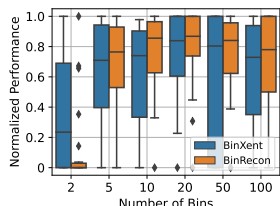

Figure 4: Empirical analysis on the dependency between the number of bins and the downstream task performance.

result is not surprising, as utilizing too few bins can eliminate necessary information while utilizing too many bins can diminish the benefits of binning. However, a relatively strong dependency of $\rho^2 = 0.34$ and $\tau = 0.60$ was observed in a subset of examples: regression tasks with BinRecon loss with fewer than 100 bins. These findings point to the possibility that identifying the optimal number of bins for specific downstream tasks or datasets can be an intriguing topic for future research.

## 6.3 BIN INFORMATION IS NOT USABLE UNLESS IT IS PROVIDED AS A PRETEXT TASK

So far, we found that bin information is critical for achieving superior representations across various tabular data problems. However, even if we do not employ bin information as an explicit pretext task, it remains accessible from the raw values. In this section, we evaluate how accurately the learned representations can predict bin indices when we optimize ValueRecon or MaskXent during SSL. To gauge this, we measure the relative error increase against the results of BinRecon case. As shown in Table 9 in the supplementary material, the prediction error is steeply increased at an average of 66.3% when bin information is not provided. This underscores that while bin information can be derived from the data, its utility is markedly compromised unless it is adopted as a pretext task.

## 7 CONCLUSION

In this work, we suggest a novel pretext task based on binning which can manipulate the unique properties of tabular datasets. The binning task can effectively address the challenges in tabular SSL, including mitigating the feature heterogeneity and learning the irregularities. Importantly, our method focuses exclusively on modifying the objective function and is independent of specific architectures or augmentation methods. Based on the extensive experiments, we found that the binning task not only consistently improves the unsupervised representation learning but also is beneficial to providing good initial weights for fine-tuning. In this study, we've uncovered the potential of leveraging the inherent properties of tabular data as pretext tasks for SSL. However, many unique characteristics remain unexplored, such as hierarchical relationships between features. We hope our work inspires further investigations into tabular-data-specific SSL in the future.

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
