# OpenReview forum: "Binning as a Pretext Task: Improving Self-Supervised Learning in Tabular Domains"
_ICLR.cc/2024/Conference — Submitted to ICLR 2024_

### Official Review · Reviewer_nMrZ · 2023-10-28

**Soundness:** 3 good
**Presentation:** 3 good
**Contribution:** 3 good
**Rating:** 8
**Confidence:** 3

**Summary:**

This paper introduces a novel method for self-supervised learning (SSL) in tabular data domains using a pretext task based on the classical binning method. Instead of reconstructing raw values, the model is trained to reconstruct bin indices, which offers a unified approach to handle both categorical and numerical features. This method equips the encoder with the ability to handle irregularities typical in tabular data and standardizes features to alleviate discrepancies. Empirical evaluations across 25 public datasets demonstrated consistent improvements in representation learning performance for various downstream tasks. The binning technique not only improves unsupervised learning results but also offers effective pretraining strategies.

**Strengths:**

Introduction of a Novel Pretext Task: The paper introduces a unique approach for self-supervised learning in tabular data using the classical binning method. By focusing on reconstructing bin indices instead of raw values, it offers a new perspective in SSL for tabular datasets.

Addressing Data Heterogeneity: The binning approach efficiently handles the inherent heterogeneity of tabular data (comprising both categorical and numerical features). It ensures all features are standardized, preventing any particular feature from dominating the learning process.

Well-written: The paper presents its ideas and findings in a clear, organized, and articulate manner, making it accessible and informative.

**Weaknesses:**

W1 Performance:

While the proposed framework works well in SSL settings, the performance in supervised settings is relatively marginal. Achieving the best results on MNIST, which is not purely tabular data, is less persuasive.

W2 Understanding the model's performance:

In SSL and supervised settings, there is limited theory analysis, discussion, or visualization to illustrate why the proposed method is effective or suboptimal. This is important as it might be critical for pushing the research of tabular data further.

**Questions:**

The main questions are listed in Weaknesses. I'd raise my score if they were appropriately addressed.


Could the authors remind me of the difference between masking with random value in your paper and the previous pretext task of estimating mask vectors from corrupted tabular data? If it is not the same, could the authors compare it in your experiments?[1]


[1] VIME: Extending the Success of Self- and Semi-supervised Learning to Tabular Domain

---

> ### Author Response · Authors · 2023-11-14
> **Response to Weakness-1 (Performance) (1)**
>
> (As the same point has been questioned by Reviewer wzCM, we respond with the same explanation as follows.)
>
> In this study, we introduce the binning loss to learn the representations that can capture the irregular functions similar to tree-based methods in the absence of supervised information.
>
> To keep our methodology in an unsupervised setup and to validate the benefits of introducing a new loss function without other influences from the architecture or optimization changes, we intentionally exclude the additional tuning techniques that need supervised information. Please consider how strange to find the best-fitted deep architectures and training details for each dataset based on “supervised” performance to examine the proposed “unsupervised” learning framework. If we allow searching for the data-dependent hyperparameter tuning in addition to the SSL-related ones (corresponding to input transformation and SSL objectives), the resulting performance gain cannot be directly attributed to the new SSL method, and the interpretation should be highly misleading.
>
> However, all the baselines in Table 3 utilize the severe hyperparameter search related to the deep architectures (e.g., depth, width, attention blocks) and the optimization (e.g., regularization term, training epochs, optimizer). It means they select the optimal hyperparameters by monitoring the supervised performance for each dataset.
>
> The hyperparameter setup is critical in determining the downstream task performance in most DNN approaches. For example, when we reproduce the T2G-Former backbone model trained with the supervised loss from scratch with the default configurations in the official implementation, we achieve the downstream task accuracy of 0.829 for CH dataset and 0.843 for AD dataset, while the performances after hyperparameter tuning correspond to 0.863 and 0.860, respectively. These results indicate that the hyperparameter tuning can hinder the clear interpretation of experimental results.
>
> Even though the hyperparameter search is essential for tabular learning performance, in this study, we did not implement any hyperparameter search related to the deep architecture or optimization, as explained in Section 5 and supplementary material. We train identical deep networks for each dataset following the same optimization protocols while only the pretext task is different.
>
> This experimental setup makes us examine the effect of changing pretext tasks only, without the effect of other design parameters. So, the performance improvements in Table 1 and Table 2 support the pure benefits of binning loss against the different objective functions.
>
> From this perspective, Table 3 provides somewhat unfair comparisons. (Note that ours with [Supervised=Yes] refers to the fine-tuned models for the fixed architectural and optimization-related hyperparameters.) Nevertheless, our method outperforms all the other baselines for 4 datasets among 12 datasets in Table 3.

---

> ### Author Response · Authors · 2023-11-14
> **Response to Weakness-1 (Performance) (2)**
>
> (As the same point has been questioned by Reviewer wzCM, we respond with the same explanation as follows.)
>
> To enhance the interpretability of our study, we will revise our manuscript as follows.
>
> (1) Ensembling the predictions
>
> Tree-based methods have many advantages for tabular data problems in addition to capturing the irregularities. The performance lag behind tree-based methods should be explained as we only aimed to resemble one specific ability of tree-based methods to capture the irregularities. One of the key factors for the superior performance of tree-based methods is making a prediction based on ensembles instead of a single tree. [1, 2] found that ensembling DNNs also provides additional performance gain for various tabular data problems. In revision, we will include the ensemble results to improve the downstream task performance.
>
> [1] Y Gorishniy et al., On Embeddings for Numerical Features in Tabular Deep Learning, NeurIPS, 2022.
>
> [2] S Badirli et al., Gradient Boosting Neural Networks: GrowNet,  arXiv, 2020.
>
> (2) Utilizing more powerful backbone models
>
> In Table 1 and 2, the binning loss consistently improves the downstream task performance for a given backbone architecture. We often find large performance gaps when we compare the supervised performance (Baseline-2) between MLPs, FT-Transformers, and other recent models in Table 3. Based on the consistent performance improvement of ours against the supervised counterparts for MLPs and FT-Transformers, we expect that the binning loss also can make an additional improvement with the better backbone models like T2G-Former. We are now experimenting on the T2G-Former with the official implementation codes and the default setups, and we will present the results in Table 3 in the revised version.
>
> (3) Providing more results with the fixed hyperparameters
>
> We agree that Table 3 can provide wrong insights because of the different training hyperparameters. To alleviate this issue, we will provide the additional results in Table 2 with the baselines, fine-tuned from the pre-trained weights on the other pretext tasks listed in Table 1. In this case, the hyperparameters are fixed and only the pretraining objectives are different, so we can attribute the performance gain of our method to changing the pretext task as reconstructing the bin indices instead of the raw values. We already have the full results, and we found that the binning loss consistently outperforms the others. We will add the results in Table 2.
>
> (4) Additional warning to interpret the results in Table 3 (Section 5.3)
>
> As we denoted in the manuscript, “we directly reference the reported performances in the papers, so the model capacity and the optimization strategies would be different” for the baseline methods. In this section, we intend to explain how our method, including the minimal change of objective function during pretraining, is powerful even compared to the heavily tuned supervised methods. However, we acknowledge the reviewer’s point that this result could cause some misunderstandings. Thus, we will add an explanation about how the comparison can be unfair and what our intention is in Section 5.3.
>
> We will reflect the above items in the revised manuscript. Thanks for the feedback.

---

> ### Author Response · Authors · 2023-11-14
> **Response to Weakness-2 and Question-1**
>
> ### Weakness-2
>
> (As the same point has been questioned by Reviewer wzCM, we respond with the same explanation as follows.)
>
> Thanks for the suggestion. We agree that the visualization analysis can improve the interpretability of our results.
>
> In the manuscript, we provide some discussion based on the experimental results related to the benefits of binning as a pretext task in Section 6. In summary, we found (1) grouping similar values is the most critical factor for the successful implementation of binning (Section 6.1), and (2) bin information is not usable unless it is provided as a pretext task (Section 6.3). To enhance the interpretability of our analysis, we will add the visualization results in the revision.
>
> To show that the binning loss can effectively encourage grouping factor, i.e., to make the similar values (i.e., in the same bins) closer to each other and separated from other values, we visualize the representation vectors after SSL with the different pretext task of ValueRecon and BinRecon. Because the representation vectors are high-dimensional, we implement PCA (or t-SNE) for better interpretability. As a result, we found that the representation vectors are grouped only when we utilize the BinRecon loss. These visualization results will be added in Section 6.1 to enhance the effect of grouping for the binning task. Thanks for the suggestion.
>
> ### Question-1
>
> We will provide a more detailed description of the difference between masking with random values and the pretext task of estimating masking vectors. In brief, the former corresponds to the input transformation, and the latter corresponds to the SSL objective.
> (1) “Masking with random value” refers to an input transformation function. We provide the description in Section 3(Input transformation) and Figure 2(b) in the manuscript. This process involves altering the input data by replacing certain values with random ones, without any specific instruction related to learning from these transformations.
>
> (2) The previous pretext task of “estimating mask vectors” from corrupted tabular data refers to a self-supervised objective. We provide the description in Section 3(SSL objectives) and equation $\mathcal{L}_\text{MaskXent}$. It aims to predict which features have been masked during the corruption process, and the corruption process includes masking as constant and masking as random.
>
> VIME [3] consists of the input transformation of masking as random and the SSL objectives of the sum of MaskXent and ValueRecon. In Table 1, the rows with (Masking=TRUE, Replacing value=Random, SSL Objectives=MaskXent+ValueRecon) correspond to the VIME approach. In addition, we compare all the possible combinations suggested in [3], allowing for a comprehensive understanding.
>
> We hope this explanation provides a clearer understanding of the differences between these two concepts and their respective roles in our methodology.
>
> [3] J Yoon et al., VIME: Extending the Success of Self- and Semi-supervised Learning to Tabular Domain, NeurIPS, 2020.

---

> ### Author Response · Authors · 2023-11-18
> **Supplementary material updates as a response to Weakness-1,2**
>
> We attached the additional results and visualization analysis in Section E in the supplementary material. We aim to incorporate these experimental results in either the main text or the supplemental materials of the final manuscript. Due to page limitations, we might need to implement some modifications to include all the necessary content efficiently. Please refer to the revised supplementary material for the preliminary results. Thanks for the suggestion.

---

> > ### Comment · Reviewer_nMrZ · 2023-11-22
> >
> > I appreciate the authors' reply. I've also read other reviewers' comments. I'd raise the score from my side, given the novel pretask.

---

> > > ### Author Response · Authors · 2023-11-22
> > >
> > > Thanks for your feedback.

---

### Official Review · Reviewer_j3rL · 2023-10-30

**Soundness:** 3 good
**Presentation:** 1 poor
**Contribution:** 2 fair
**Rating:** 5
**Confidence:** 4

**Summary:**

This paper proposes a new pretext task in pretext-task-based self-supervision that predicts the bin index or value of a given example  quantised/binned into intervals. The work closely follows the philosophy of VIME, with the pretext task being thought of an encoder, and the downstream classification / regression as the decoder. The method evaluates itself against 8 benchmark methods over 12 datasets (Tables 2-3), and performs ablation over 3 components of the binning operation, and offers reasoning over hyperparameters e.g. the number of bins.

**Strengths:**

The  idea of binning the ranges of input variables is novel, and seems to have been developed in the spirit of course-graining these ranges, something that tree-based supervised models seem to like, thereby bringing that courseness as an inductive bias from trees as Gorshiniy'22 suggests is useful. The presented evaluations cover both quantification axes. Those evaluations have been performeed on both binary and multi-class classification, and regression.

**Weaknesses:**

The connection to Gorshiniy'22 's idea seems hypothetical at best, and does not convince that quantising has the desired causality.

The paper goes into VIME's masking method as if it was proposed here. The masking method is indeed used in conjugation with binning, but  may be de-stressed here.

**Questions:**

In the ablation study, shouldn't the absence of grouping have beenc investigated, rather than its presence? Isn't the baseline the one with all three configurations True?

How would course-graining compare against a method like SSS'23 (Syed and Mirza) that assumes smoothness but adds jumps in discrete sub-Gaussian steps?

---

> ### Author Response · Authors · 2023-11-14
> **Response to Reviewer j3rL**
>
> ### Weakness-1
>
> In [Gorshiniy'22], quantization(binning) is introduced as an activation function between layers. So, they quantized the representation vectors instead of the input samples. In our approach, however, we directly quantized the input samples themselves and used them as the labels for the pretext task. Thus, our approach is quite different from [Gorshiniy'22].
>
> One important hypothesis underlying our method is that tabular datasets can be effectively represented through irregular functions. To validate our hypothesis, we have conducted comprehensive experiments by changing the SSL objectives only. Other options, including the architectural and optimization-related hyperparameters, were held constant for all experiments. This experimental setup allows us to attribute differences in representation (or pretrained weights) quality exclusively to the choice of SSL objectives. The consistently superior performance of binning loss against the others substantiates that quantizing(binning) should provide the desired property for various tabular data problems.
>
> However, we do not claim the causality that quantizing is always beneficial for tabular learning. Optimization of deep networks is quite complicated to explain based on the elegant theoretical backgrounds. Even though we conducted strictly controlled experiments with various datasets, the learning dynamics is too complex to understand and confirm the solid causality. Thus, we agree that quantizing is not a desired “causality” for all cases, and we only empirically found that quantizing is helpful for tabular SSL.
>
> ### Weakness-2
>
> Thank you for pointing that out. We would add the comments to stress the importance of masking methods in Section 4.
>
> ### Question-1
>
> We will explain more detailed experimental setups of the ablation study in Section 6.2. First, we set the baseline as optimizing the original BinRecon loss, wherein all three configurations are set to TRUE (Yes in Table 4). This setup enhances the interpretability by measuring the relative performance drop against the baseline.
>
> To investigate the impact of each factor, we systemically excluded each factor and reported the performance in different rows in Table 4. This approach aligns with your understanding: we indeed examine the absence rather than the presence of individual factors.
>
> In the case of the grouping factor, the standardizing factor cannot be satisfied due to the unique values of the features varying significantly. Thus, in this study, both factors had to be omitted simultaneously, as presented in the 4th row in Table 4. By comparing the results in the 3rd and the 4th rows of Table 4, we conclude that the absence of the grouping factor results in the most significant performance drop.
>
> This analysis enables us to better understand the distinct contributions of each factor to the overall performance.
>
> ### Question-2
>
> Thanks for the question. However, we cannot find the paper with the keyword of SSS'23 (Syed and Mirza). Could you give me the complete title of the paper?

---

### Official Review · Reviewer_m34i · 2023-10-31

**Soundness:** 3 good
**Presentation:** 3 good
**Contribution:** 3 good
**Rating:** 6
**Confidence:** 4

**Summary:**

This paper presents a pretext task aimed at enhancing self-supervised learning within tabular domains. The core concept involves discretizing numerical features through binning and encoding categorical features with one-hot vectors. The experiments conducted demonstrate that this pretext task leads to improved feature representations and more effective weightings for downstream tasks.

**Strengths:**

- The method is comprehensible, and the illustrations are clear.
- The binning solution is intriguing, and this pretext task can be seamlessly integrated with numerous modifications of SSL methods.
- The paper effectively demonstrates that the most crucial aspect of this pretext task lies in grouping similar values.

**Weaknesses:**

- The number of bins significantly impacts the performance of the representation, but it can be challenging to predefine.
- It appears that this method primarily enhances a single network's performance on a specific dataset. What about neural networks trained on multiple heterogeneous tabular datasets?

**Questions:**

- What is the recommended approach for defining the hyperparameter related to the number of bins?
- Lately, several works like "transtab" have emerged, aiming to generate a single neural network for multiple heterogeneous datasets. Can the method proposed in this paper contribute to enhancing the representation of a global tabular network across various datasets with differing feature spaces?

---

> ### Author Response · Authors · 2023-11-14
> **Response to Reviewer m34i**
>
> ### Weakness-1 and Question-1
>
> As explained in Section 6.2, we found no clear relationship between the number of bins and the downstream task performance. (This result is not surprising, as utilizing too few bins can eliminate necessary information while utilizing too many bins can diminish the benefits of binning.)
>
> The results parallel the choice of data augmentation in self-supervised learning within vision and NLP domains. In most domains, we follow a rule-of-thumb protocol that compares linear evaluation performance on a validation dataset. In this study, we adapted this method to identify the (sub)optimal hyperparameter for the number of bins.
>
> We acknowledge the potential benefits of developing a more systematic method to predefine the number of bins in an unsupervised setup. This topic presents an intriguing avenue for future research and should significantly enhance our methodology. We suggest beginning this exploration with insights from prior studies on finding optimal data augmentation methods in unsupervised setups in vision domains [1, 2, 3]. We thank the reviewer for this valuable suggestion and agree that this discussion will largely help in understanding our method. We will include this discussion in Section 6.2 of the manuscript.
>
> [1] Y Tian et al., What Makes for Good Views for Contrastive Learning?, NeurIPS, 2020.
> [2] Q Xie et al., Unsupervised Data Augmentation for Consistency Training, NeurIPS, 2020.
> [3] D Berthelot et al., ReMixMatch: Semi-Supervised Learning with Distribution Alignment and Augmentation Anchoring, ICLR, 2020.
>
>
> ### Weakness-2 and Question-2
>
> We anticipate that our method could be beneficial in developing a unified network for multiple heterogeneous datasets with certain modifications. Because our method is quite simple as changing the objective function only during pretraining stage, our method can be easily combined with a variety of deep architectures and applications. Using multiple tabular datasets presents an intriguing topic for future research, and we propose the following detailed suggestions. These suggestions could significantly enhance the versatility of our method in the context of unified networks for heterogeneous datasets.
>
> (1) Adapting the additional embedding modules: Our current framework does not support heterogeneous tabular datasets with different feature dimensions. By incorporating additional embedding modules like a feature tokenizer [4], it can be feasible to train the shared backbone encoder with multiple heterogeneous datasets based on the binning loss in a unified manner.
>
> (2) Semi-supervised learning framework: When we change the training framework from unsupervised to semi-supervised learning, the binning loss could serve as a regularization loss for unlabeled datasets. This method allows the labeled and unlabeled datasets to belong to different data domains.
>
> (2) Prompts with Large Language Models (LLMs): The binning task is model-agnostic and could be applied to LLMs. By utilizing prompts to indicate the specific dataset in use, LLMs can be the powerful backbone for training with heterogeneous datasets.
>
> [4] B Zhu et al., XTab: Cross-table Pretraining for Tabular Transformers, ICML, 2023.

---

> > ### Comment · Reviewer_m34i · 2023-11-23
> >
> > Thank you for the author's response. It is plausible that the proposed method could find application in constructing a global neural network for heterogeneous tabular tasks. This method serves as a general pre-task that has the potential to enhance the performance of tabular networks.

---

### Official Review · Reviewer_wzCM · 2023-11-01

**Soundness:** 2 fair
**Presentation:** 2 fair
**Contribution:** 2 fair
**Rating:** 5
**Confidence:** 4

**Summary:**

This study aims to enhance the representation learning capabilities of deep networks for managing the heterogeneous features present in tabular data. Building upon previous insights in the field of tabular learning, this research highlights the challenge faced by deep networks in modeling irregular functions. To address this issue, the paper introduces an innovative self-supervised pretext task based on the classical data mining method of binning. Unlike previous methods that focused on reconstructing the original tabular cell values, this approach attempts to reconstruct bin indices. By doing so, the encoder captures irregular dependencies and mitigates feature heterogeneity. The study concludes with extensive empirical experiments conducted across various tabular datasets, demonstrating the method's efficacy in enhancing tabular representation learning performance.

**Strengths:**

- The paper offers a fresh perspective building upon prior research [1]. The proposed solution is not only reasonable but also simplified, ensuring compatibility with various encoder architectures.
 - This paper is well-crafted, providing a precise and clear description of the method employed.
 - The paper meticulously outlines the experimental settings, conducting the experiments ten times with different random seeds and presenting standard deviations. This rigorous approach enhances the paper's credibility.
 - The code implementation is exemplary, featuring a well-organized structure that is easy to follow. It includes comprehensive parameters, pre-trained weights, and detailed training logs, making it highly reproducible.
  - Remarkably, the method exhibits efficiency, requiring minimal computational resources, specifically just a single NVIDIA GeForce RTX3090.

[1] Grinsztajn L, Oyallon E, Varoquaux G. Why do tree-based models still outperform deep learning on typical tabular data?[J]. Advances in Neural Information Processing Systems, 2022, 35: 507-520.

**Weaknesses:**

- The primary limitation of this work lies in its experimental results. Table 3 illustrates that this method still falls short of achieving comparable performance to the tree-based method even though they claim this method can mitigate the impact of feature heterogeneity. Additionally, the method's performance lags behind that of state-of-the-art neural network-based models such as T2G-Former.
- The pretext task has only been tested on TRUE. It is advisable to diversify the experiment results by trying different backbone models to ensure the robustness and versatility of the proposed approach.
- The paper overlooks significant related works, such as TabGSL [2], a work focused on supervised tabular prediction, as well as Tabular Language Models (TaLMs), TableFormer [3], and TABBIE[4]. Incorporating these relevant studies would provide a more comprehensive understanding of the research landscape.
- It is recommended to incorporate interpretable analysis methods. For example, visualizing decision boundaries could validate whether binning as a pretext task effectively fits irregular functions, enhancing the paper's explanatory power.

[2] Liao J C, Li C T. TabGSL: Graph Structure Learning for Tabular Data Prediction[J]. arXiv preprint arXiv:2305.15843, 2023.

[3] Yang J, Gupta A, Upadhyay S, et al. TableFormer: Robust transformer modeling for table-text encoding[J]. arXiv preprint arXiv:2203.00274, 2022.

[4] Iida H, Thai D, Manjunatha V, et al. Tabbie: Pretrained representations of tabular data[J]. arXiv preprint arXiv:2105.02584, 2021.

**Questions:**

- Why does the experimental performance lag behind tree-based methods, recap the motivation for this method is to enable neural networks to handle irregular functions akin to tree-based methods.
- Why does the performance of this method lag behind the SoTA NN-based methods?
- How long does this method take to pre-train on a single NVIDIA GPU?
- How about the transfer ability for bining pretext task? Can it achieve zero-shot or few-shot learning?
- How do self-supervised learning methods compare to methods based on large language models?

---

> ### Author Response · Authors · 2023-11-14
> **Response to Weakness-1, Question-1, and Question-2 (1)**
>
> (As the same point has been questioned by Reviewer nMrZ, we respond with the same explanation as follows.)
>
> In this study, we introduce the binning loss to learn the representations that can capture the irregular functions similar to tree-based methods in the absence of supervised information.
>
> To keep our methodology in an unsupervised setup and to validate the benefits of introducing a new loss function without other influences from the architecture or optimization changes, we intentionally exclude the additional tuning techniques that need supervised information. Please consider how strange to find the best-fitted deep architectures and training details for each dataset based on “supervised” performance to examine the proposed “unsupervised” learning framework. If we allow searching for the data-dependent hyperparameter tuning in addition to the SSL-related ones (corresponding to input transformation and SSL objectives), the resulting performance gain cannot be directly attributed to the new SSL method, and the interpretation should be highly misleading.
>
> However, all the baselines in Table 3 utilize the severe hyperparameter search related to the deep architectures (e.g., depth, width, attention blocks) and the optimization (e.g., regularization term, training epochs, optimizer). It means they select the optimal hyperparameters by monitoring the supervised performance for each dataset.
>
> The hyperparameter setup is critical in determining the downstream task performance in most DNN approaches. For example, when we reproduce the T2G-Former backbone model trained with the supervised loss from scratch with the default configurations in the official implementation, we achieve the downstream task accuracy of 0.829 for CH dataset and 0.843 for AD dataset, while the performances after hyperparameter tuning correspond to 0.863 and 0.860, respectively. These results indicate that the hyperparameter tuning can hinder the clear interpretation of experimental results.
>
> Even though the hyperparameter search is essential for tabular learning performance, in this study, we did not implement any hyperparameter search related to the deep architecture or optimization, as explained in Section 5 and supplementary material. We train identical deep networks for each dataset following the same optimization protocols while only the pretext task is different.
>
> This experimental setup makes us examine the effect of changing pretext tasks only, without the effect of other design parameters. So, the performance improvements in Table 1 and Table 2 support the pure benefits of binning loss against the different objective functions.
>
> From this perspective, Table 3 provides somewhat unfair comparisons. (Note that ours with [Supervised=Yes] refers to the fine-tuned models for the fixed architectural and optimization-related hyperparameters.) Nevertheless, our method outperforms all the other baselines for 4 datasets among 12 datasets in Table 3.

---

> ### Author Response · Authors · 2023-11-14
> **Response to Weakness-1, Question-1, and Question-2 (2)**
>
> (As the same point has been questioned by Reviewer nMrZ, we respond with the same explanation as follows.)
>
> To enhance the interpretability of our study, we will revise our manuscript as follows.
>
> (1) Ensembling the predictions
>
> Tree-based methods have many advantages for tabular data problems in addition to capturing the irregularities. The performance lag behind tree-based methods should be explained as we only aimed to resemble one specific ability of tree-based methods to capture the irregularities. One of the key factors for the superior performance of tree-based methods is making a prediction based on ensembles instead of a single tree. [1, 2] found that ensembling DNNs also provides additional performance gain for various tabular data problems. In revision, we will include the ensemble results to improve the downstream task performance.
>
> [1] Y Gorishniy et al., On Embeddings for Numerical Features in Tabular Deep Learning, NeurIPS, 2022.
>
> [2] S Badirli et al., Gradient Boosting Neural Networks: GrowNet,  arXiv, 2020.
>
> (2) Utilizing more powerful backbone models
>
> In Table 1 and 2, the binning loss consistently improves the downstream task performance for a given backbone architecture. We often find large performance gaps when we compare the supervised performance (Baseline-2) between MLPs, FT-Transformers, and other recent models in Table 3. Based on the consistent performance improvement of ours against the supervised counterparts for MLPs and FT-Transformers, we expect that the binning loss also can make an additional improvement with the better backbone models like T2G-Former. We are now experimenting on the T2G-Former with the official implementation codes and the default setups, and we will present the results in Table 3 in the revised version.
>
> (3) Providing more results with the fixed hyperparameters
>
> We agree that Table 3 can provide wrong insights because of the different training hyperparameters. To alleviate this issue, we will provide the additional results in Table 2 with the baselines, fine-tuned from the pre-trained weights on the other pretext tasks listed in Table 1. In this case, the hyperparameters are fixed and only the pretraining objectives are different, so we can attribute the performance gain of our method to changing the pretext task as reconstructing the bin indices instead of the raw values. We already have the full results, and we found that the binning loss consistently outperforms the others. We will add the results in Table 2.
>
> (4) Additional warning to interpret the results in Table 3 (Section 5.3)
>
> As we denoted in the manuscript, “we directly reference the reported performances in the papers, so the model capacity and the optimization strategies would be different” for the baseline methods. In this section, we intend to explain how our method, including the minimal change of objective function during pretraining, is powerful even compared to the heavily tuned supervised methods. However, we acknowledge the reviewer’s point that this result could cause some misunderstandings. Thus, we will add an explanation about how the comparison can be unfair and what our intention is in Section 5.3.
>
> We will reflect the above items in the revised manuscript. Thanks for the feedback.

---

> ### Author Response · Authors · 2023-11-14
> **Response to Weakness-2,3,4**
>
> ### Weakness-2
> Thanks for your suggestion regarding the versatility of our approach. We agree that assessing various backbone encoders can largely enhance the robustness and versatility of our binning method.
>
> As you rightly pointed out, our method is model-agnostic, allowing flexibility in choosing backbone encoders. In line with your recommendation, we have expanded our experiments to include the ResNet and T2G-Former model, in addition to our initial tests with MLPs and FT-Transformers. We chose the ResNet due to its distinctly different architectural design compared to MLPs and FT-Transformers, and T2G-Former due to its superior performance.
>
> We provide the partial results in the same format as Table 2 in the manuscript as follows. We found that the binning task is effective with ResNet and T2G-Former backbone as well. These results further validate the adaptability and effectiveness of our method across diverse DNN architectures.
>
> * Partial results of ResNet [Dataset / Supervised (Baseline-2) / Binning (Ours-2)]: [Classification] CH/0.827/0.839, PO/0.795/0.801, GE/0.484/0.496, [Regression] CA/0.706/0.650, HO/4.004/3.936, FI/10226.508/9800.832
>
> * Partial results of T2G-Former [Dataset / Supervised (Baseline-2) / Binning (Ours-2)]: [Classification] CH/0.829/0.837, PO/0.881/0.892, GE/0.681/0.718, [Regression] CA/0.475/0.474, HO/3.269/3.224, FI/10750.850/10660.164
>
> We will add the full results in Table 2 in the final manuscript. We hope these additional experiments will adequately address the concerns about the robustness and versatility of our proposed approach.
>
> ### Weakness-3
>
> Thanks for the suggestion. We agree that a more comprehensive review of the literature on tabular deep learning methods is essential for our paper. In response to your suggestion, we will revise the related works section to include a broader range of studies, specifically addressing the significant contributions of works like TabGSL [2], Tabular Language Models (TaLMs), TableFormer [3], and TABBIE[4].
>
> This expanded review will be situated in the "Tabular Deep Learning" subsection in Section 2, preceding the sentence "In this paper, …". By incorporating these relevant studies, we aim to provide a complete understanding of the current research landscape in this field, thereby situating our work within the broader context of recent advancements in tabular data analysis.
>
> ### Weakness-4
>
> Thanks for the suggestion. We agree that the visualization analysis can improve the interpretability of our results.
>
> In the manuscript, we provide some discussion based on the experimental results related to the benefits of binning as a pretext task in Section 6. In summary, we found (1) grouping similar values is the most critical factor for the successful implementation of binning (Section 6.1), and (2) bin information is not usable unless it is provided as a pretext task (Section 6.3). To enhance the interpretability of our analysis, we will add the visualization results in the revision.
>
> To show that the binning loss can effectively encourage grouping factor, i.e., to make the similar values (i.e., in the same bins) closer to each other and separated from other values, we visualize the representation vectors after SSL with the different pretext task of ValueRecon and BinRecon. Because the representation vectors are high-dimensional, we implement PCA (or t-SNE) for better interpretability. As a result, we found that the representation vectors are grouped only when we utilize the BinRecon loss. These visualization results will be added in Section 6.1 to enhance the effect of grouping for the binning task. Thanks for the suggestion.

---

> ### Author Response · Authors · 2023-11-14
> **Response to Question-3,4,5**
>
> ### Question-3
>
> The pre-training time varies depending on several factors, including the dataset size, the input transformation functions, and the backbone architectures. We provide the approximated training time for two examples as follows. The approximations are obtained using a single NVIDIA GPU.
>
> (1) CH dataset: This dataset consists of 6400 training samples and 10 features. The approximate pre-training time with BinRecon loss is as follows: [MLP with no input transformation] 1 minute; [MLP with input transformation] 1 minute; [FT-Transformer with no input transformation] 7 minutes; and [FT-Transformer with input transformation] 7 minutes. These results are reproducible in codes in supplementary material.
>
> (2) MI dataset: This dataset is the largest dataset among the 25 datasets used in this study, and it consists of 723k training samples with 136 features. The approximate pre-training time with BinRecon loss is as follows: [MLP with no input transformation] 40 minutes; [MLP with input transformation] 65 minutes; [FT-Transformer with no input transformation] 150 minutes; and [FT-Transformer with input transformation] 175 minutes.
>
> ### Question-4
>
> In this study, we have focused primarily on training and testing within in-distribution datasets. Nonetheless, we recognize the potential of our method in broader applications, including in scenarios with heterogeneous datasets, like zero-shot or few-shot learning. Because our method is quite simple as changing the objective function only during pretraining stage, our method can be easily combined with a variety of deep architectures and applications. Transfer learning with binning task also can be an intriguing topic for future research, and we propose the following detailed suggestions. These suggestions could significantly enhance the versatility of our method in the context of transfer learning.
>
> (1) Adapting the additional embedding modules: Our current framework does not support heterogeneous tabular datasets with different feature dimensions. By incorporating additional embedding modules like a feature tokenizer [3], it can be feasible to train the shared backbone encoder with multiple heterogeneous datasets based on the binning loss in a unified manner.
>
> (2) Prompts with Large Language Models (LLMs): The binning task is model-agnostic and could be applied to LLMs. By utilizing prompts to indicate the specific dataset in use, LLMs can be the powerful backbone for training with heterogeneous datasets.
>
> (3) Defining self-generated tasks by separating all the single features: Inspired by the recent study [4], which significantly improved tabular learning performance in transfer learning contexts, we could explore self-generating diverse few-shot tasks. This would involve treating randomly chosen columns as target labels. When we adapt our methodology with [4], we could use bin indices as the target labels during meta-learning. However, the previous approach considers the classification tasks only, and it is not directly compatible with regression tasks, necessitating further exploration.
>
> [3] B Zhu et al., XTab: Cross-table Pretraining for Tabular Transformers, ICML, 2023.
>
> [4] J Nam et al., STUNT: Few-shot Tabular Learning with Self-generated Tasks from Unlabeled Tables, ICLR, 2023.
>
> ### Question-5
>
> Self-supervised learning is a broader concept that encompasses various techniques for training models without explicit downstream-task labels, while large language models (LLMs) are a specific class of models that leverage self-supervised learning as part of their training process to achieve impressive natural language understanding and generation capabilities. Our method, which focuses on utilizing the binning loss, fits within the realm of self-supervised learning. It is specifically designed to enhance learning from tabular data by leveraging the intrinsic patterns and distributions within the data. We recognize the potential of integrating our method with large language models, and we expect that training LLMs with a binning loss as a pretext task could be a promising direction for future research.
>
> In conclusion, while self-supervised learning methods provide a broad framework for model training without labeled data, LLMs represent a specialized application within this framework. Our method's compatibility with LLMs opens up intriguing possibilities for future explorations in this space.

---

> ### Author Response · Authors · 2023-11-18
> **Supplementary material updates as a response to Weakness-1,2, Question-1,2**
>
> We attached the additional results and visualization analysis in Section E in the supplementary material. We aim to incorporate these experimental results in either the main text or the supplemental materials of the final manuscript. Due to page limitations, we might need to implement some modifications to include all the necessary content efficiently. Please refer to the revised supplementary material for the preliminary results. Thanks for the suggestion.

---

### Author Response · Authors · 2023-11-14
**Common response to all reviewers**

Dear Reviewers,

We extend our sincere gratitude for your detailed and insightful feedback on our paper. The comments are greatly helpful in guiding our revisions. We are confident in addressing all the issues raised and we will provide thorough explanations for each weakness and question. Also, the explanations will be incorporated into the manuscript. Please feel free to raise any additional questions or concerns during the discussion period. We welcome the opportunity to further clarify and enhance our work based on the reviewers’ expert guidance.

---

### Meta-Review · Area_Chair_Xif7 · 2023-12-05

**Metareview:**

The authors propose a simple, yet effective strategy to learn latent representations of tabular data, by autoencoding the bin of each feature value. The idea in itself is a very simple and elegant self-supervised representation learning strategy, and empirically improves over placebo autoencoders that reconstruct raw feature values. The paper is well-written and nicely organized.

Unfortunately, the empirical evidence to support the state-of-the-art nature of the method is lacking. The technique does not outperform classical gradient-boosted decision tree methods for tabular datasets, which are more lightweight techniques than the proposed binning neural network. It is unclear under what circumstances might a practitioner want to use the proposed method, as opposed to simply using existing libraries for XGBoost/Catboost? Therefore, I recommend rejection.

**Justification For Why Not Higher Score:**

The experimental aspect of the work is still premature, the method is not yet competitive to gradient-boosted decision tree techniques for tabular datasets.

**Justification For Why Not Lower Score:**

N/A

---

### Decision · Program_Chairs · 2024-01-16

Reject